# Large out-of-plane spin−orbit torque in topological Weyl semimetal TaIrTe₄

Lakhan Bainsla ®[1,2] ✉, Bing Zhao ®[1], Nilamani Behera ®[3], Anamul Md. Hoque ®[1], Lars Sjöström ®[1], Anna Martinelli[4], Mahmoud Abdel-Hafiez ®[5,6], Johan Åkerman ®[3,7,8] & Saroj P. Dash ®[1,9,10] ✉

The unique electronic properties of topological quantum materials, such as protected surface states and exotic quasiparticles, can provide an out-of-plane spin-polarized current needed for external field-free magnetization switching of magnets with perpendicular magnetic anisotropy. Conventional spin−orbit torque (SOT) materials provide only an in-plane spin-polarized current, and recently explored materials with lower crystal symmetries provide very low out-of-plane spin-polarized current components, which are not suitable for energy-efficient SOT applications. Here, we demonstrate a large out-of-plane damping-like SOT at room temperature using the topological Weyl semimetal candidate TaIrTe₄ with a lower crystal symmetry. We performed spin−torque ferromagnetic resonance (STFMR) and second harmonic Hall measurements on devices based on TaIrTe₄/Ni₈₀Fe₂₀ heterostructures and observed a large out-of-plane damping-like SOT efficiency. The out-of-plane spin Hall conductivity is estimated to be $(4.05 \pm 0.23) \times 10^4$ $(\hbar/2e)$ $(\Omega m)^{-1}$, which is an order of magnitude higher than the reported values in other materials.

Spin−orbit torque (SOT), utilizing the charge-to-spin conversion in a high spin−orbit coupling material (SOM) to create magnetization dynamics in an adjacent ferromagnet (FM), is expected to provide a breakthrough for next-generation memory and logic technologies[1,2]. SOT-based memory devices have the potential to challenge the devices based on spin-transfer torque, but the use of conventional SOMs leads to moderate efficiency. Furthermore, as the component of torque lies in-plane in conventional SOMs such as Pt and Ta, they are only suitable for deterministic switching of in-plane magnets[1–4]. However, for thermally stable high-density memory technologies, the industry requires magnets with perpendicular magnetic anisotropy (PMA), where additional measures, such as magnetic field assistance, will be required for

deterministic switching. The first nontrivial requirement for a practical SOT memory technology is therefore the field-free deterministic SOT switching of FMs with PMA[2].

To achieve this, SOMs with lower crystal symmetry are needed to generate out-of-plane damping-like torque components suitable for switching PMA ferromagnets. In contrast to conventional SOT, with a torque vector perpendicular to the plane of the electron's motion and the electric field, unconventional SOT produces a tilted torque vector. Recent experiments demonstrated that van der Waals SOMs with reduced crystal symmetry, such as WTe₂ in heterostructure with FMs, allow the generation of a nontrivial current-induced spin polarization with out-of-plane SOT symmetries[5–10]. More recently, field-free SOT

¹Department of Microtechnology and Nanoscience, Chalmers University of Technology, SE-41296 Göteborg, Sweden. ²Department of Physics, Indian Institute of Technology Ropar, Rupnagar 140001 Punjab, India. ³Department of Physics, University of Gothenburg, Göteborg, SE-41296 Göteborg, Sweden. ⁴Department of Chemistry and Chemical Engineering, Chalmers University of Technology, Göteborg 41296, Sweden. ⁵Department of Applied Physics and Astronomy, University of Sharjah, P. O. Box 27272 Sharjah, United Arab Emirates. ⁶Department of Physics and Astronomy, Uppsala University, Box 516, SE-751 20 Uppsala, Sweden. ⁷Center for Science and Innovation in Spintronics, Tohoku University, 2-1-1 Katahira, Aoba-ku, Sendai 980-8577, Japan. ⁸Research Institute of Electrical Communication, Tohoku University, 2-1-1 Katahira, Aoba-ku, Sendai 980-8577, Japan. ⁹Wallenberg Initiative Materials Science for Sustainability, Department of Microtechnology and Nanoscience, Chalmers University of Technology, SE-41296 Göteborg, Sweden. ¹⁰Graphene Center, Chalmers University of Technology, SE-41296 Göteborg, Sweden. ✉e-mail: lakhan.bainsla@iitrpr.ac.in; saroj.dash@chalmers.se

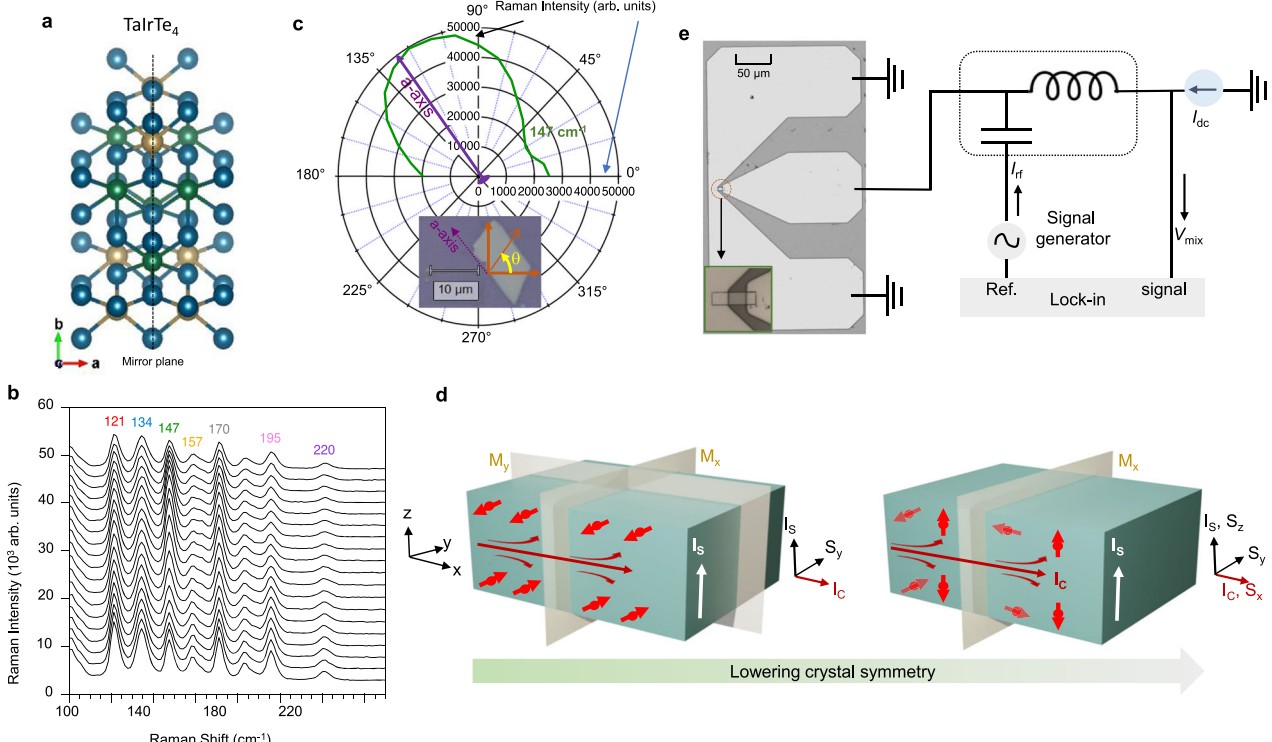

**Fig. 1 | Crystal structure, the origin of unconventional spin−orbit torque, and schematic of the spin−torque ferromagnetic resonance (STFMR) measurement set-up with device geometry.** **a** Crystal structure of $T_d$-TaIrTe$_4$ showing lower crystal symmetry with mirror plane along a-axis only. **b** Angle-dependent, polarized Raman spectra of TaIrTe$_4$ obtained by rotating the polarization of the incident laser with respect to the sample that was fixed in position. **c** The representative polar plot of the angle-dependent intensity of A$_g$ (A$_1$) mode at ~147 cm$^{-1}$ reveals a maximum value when laser polarization is aligned along the a-axis. **d** Conventional and unconventional SOT mechanisms in higher and lower crystalline symmetry

materials. In high-symmetry materials, charge current $I_C$, spin current $I_S$ and spin polarization $S$ (here $S_y$) follow the conventional orthogonality relation (1$^{st}$ diagram), while one can generate unconventional out-of-plane spin currents in lower symmetry materials where $I_C$, $I_S$ and spin polarization $S$ do not follow the orthogonality relation as spin polarization also has a z-component ($S_z$) (see 2$^{nd}$ diagram; shown by removing $M_y$ mirror plane). **e** Schematic of the spin−torque ferromagnetic resonance (STFMR) measurement set-up with TaIrTe$_4$/Ni$_{80}$Fe$_{20}$ heterostructure device geometry. Scale bar is 50 μm for the device image, inset shows the magnified view of the microbar device.

switching has been reported using WTe$_2$/Fe$_3$GeTe$_2$ heterostructures induced by out-of-plane SOT from WTe$_2$[5,7,11,12]. However, the out-of-plane SOT strengths obtained using materials such as WTe$_2$, MoTe$_2$, NbSe$_2$, Mn$_2$Au, MnPd$_3$, IrO$_2$, etc. are an order of magnitude smaller than the conventional in-plane SOT component, making it challenging to realize energy-efficient SOT devices[5–9,11–21]. Therefore, it is crucial to discover new materials that exhibit large out-of-plane spin polarization and SOT components.

The topological Weyl semimetal candidate TaIrTe$_4$ has gained significant attention as it shows the presence of bulk Weyl nodes and Fermi-arc surface states, which are unique band crossings in momentum space and useful spin textures that can give a variety of unusual electronic and charge-to-spin conversion properties[22–26]. The combination of topological spin textures and lower crystal symmetry hence makes TaIrTe$_4$ a promising candidate for energy-efficient SOT devices. Here, using spin−torque ferromagnetic resonance (STFMR) and second harmonic measurements[3,27] in TaIrTe$_4$/Ni$_{80}$Fe$_{20}$ heterostructures[3,27–29], we show a significant out-of-plane damping-like torque and a substantial out-of-plane spin Hall conductivity at room temperature.

## Results and discussion

TaIrTe$_4$ is a promising topological Weyl semimetal candidate due to its large spin−orbit coupling and broken crystal symmetry, exhibiting unique chiral spin textures of electronic bands for bulk Weyl nodes and Fermi-arc surface states[30]. TaIrTe$_4$ hosts only four type-II Weyl nodes providing the simplest model system with broken inversion symmetry.

Additionally, the lower crystal symmetry of the $T_d$-TaIrTe$_4$ structure with space group $Pmn2_1$ (Fig. 1a), can provide unconventional charge−spin conversion due to the presence of topologically non-trivial electronic states. The crystallographic alignment of the patterned devices is confirmed by performing angle-dependent, polarized Raman spectroscopy. Representative polarized Raman spectra are shown in Fig. 1b, the angle dependence is consistent with the one previously reported for $T_d$-phase TaIrTe$_4$[31]. The polar plot shown in Fig. 1c shows the integrated intensity of the A$_g$(A$_1$) Raman mode at a wavenumber of 147 cm$^{-1}$. The intensity of this vibrational mode reaches a maximum when the laser polarization is parallel to the a-axis. These properties can result in the emergence of current-induced spin polarization that does not strictly follow the orthogonal relation between the charge current ($I_C$), spin current ($I_S$), and spin polarization (S) orientation as shown in Fig. 1d. The generation of an out-of-plane spin polarization can induce an out-of-plane SOT on the adjacent ferromagnet. In contrast to conventional in-plane SOT, which applies torque parallel to the device plane, the out-of-plane SOT can more efficiently manipulate magnetic moments with perpendicular components for high-density integration.

In this study, we focus on the unconventional charge-to-spin conversion in TaIrTe$_4$, particularly the generation of out-of-plane damping-like SOT components. STFMR measurements are performed at room temperature to investigate the SOT in TaIrTe$_4$/Ni$_{80}$Fe$_{20}$ bilayers. Ni$_{80}$Fe$_{20}$ is also known as permalloy (Py). The schematic of the STFMR measurement setup is shown in Fig. 1e. The STFMR microbars were fabricated using electron-beam lithography and argon ion

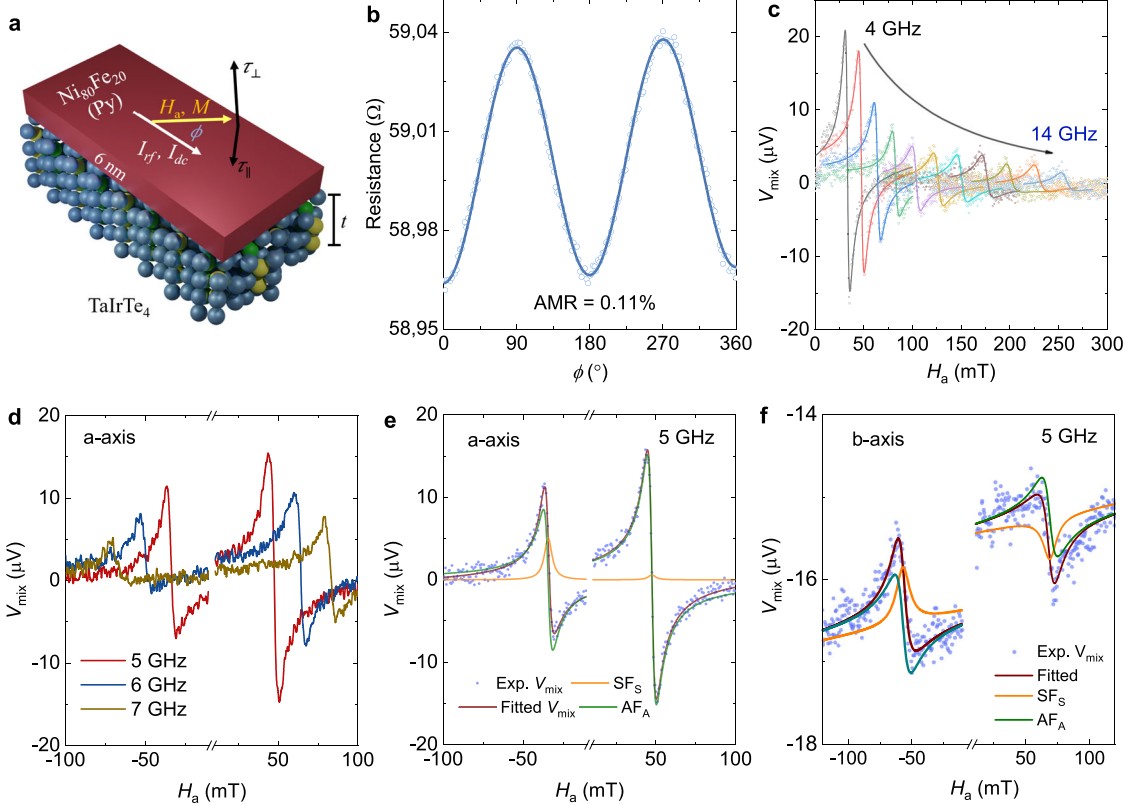

**Fig. 2 | Unconventional charge–spin conversion in TaIrTe₄. a** Schematic of the TaIrTe₄/Py heterostructure with SOT components, where $I_{rf}$ and $I_{dc}$ are the in-plane applied radio frequency and direct current, respectively. $H_a$ applied magnetic field, $M$ magnetization of the sample, and $\phi$ is the in-plane angle between applied current and magnetic field. $\tau_\parallel$ and $\tau_\perp$ are the in-plane and out-of-plane component of the damping-like torque. **b** Anisotropic magnetoresistance curve with an applied magnetic field of 100 mT and *dc* current of 0.6 mA. **c** Frequency-dependent STFMR spectra with in-plane magnetic field angle $\phi = 40°$ for a frequency range of 4-14 GHz. **d**, STFMR spectra with positive ($\phi = 40°$) and negative ($\phi = 220°$) applied

magnetic field in the frequency range of 5–7 GHz. **e, f** The experimental STFMR curves (Exp. V_mix), fitted curves using Eq. (1) (fitted V_mix), symmetric (SF_S) and antisymmetric (AF_A) contributions in the V_mix (fitted) at 5 GHz for devices fabricated along a and b-axis respectively. In **c, e**, and **f** solid symbols represent the experimental STFMR data and solid lines are fit to the obtained data using Eq. (1). For **b–e** measurements are performed in the TaIrTe₄(133 nm)/Py(6 nm) device fabricated along a-axis, while for **f** measurements are performed in TaIrTe₄(30 nm)/Py(6 nm) device fabricated along b-axis.

milling. The devices were fabricated along the longer axis of the TaIrTe₄ flakes, which is the a-axis in this class of materials[9,22,26].

In STFMR measurements, an in-plane radio frequency current $I_{rf}$ is applied along the a-axis of TaIrTe₄ while an in-plane magnetic field $H_a$ is applied at an angle $\phi$ concerning the $I_{rf}$, as shown in Fig. 2a. $I_{rf}$ in TaIrTe₄ generates a spin current in the z-direction, which is injected into the adjacent Py layer and excites the Py into a processional motion. Thanks to its anisotropic magnetoresistance (AMR), the resistance of Py oscillates with the same frequency as that of $I_{rf}$, and produces a *dc* mixing voltage V_mix, which is then measured using a lock-in amplifier. AMR is measured for a TaIrTe₄(133 nm)/Py(6 nm) device, and a value of 0.11% is obtained as shown in Fig. 2b and in Supplementary Fig. 2 for other devices. Figure 2c shows the representative STFMR signals V_mix for the TaIrTe₄(133 nm)/Py(6 nm) device at room temperature. The obtained V_mix signal is then fitted using the equation[3,32],

$$V_{mix} = SF_S(H_a) + AF_A(H_a) \qquad (1)$$

where, $F_S(H_a) = \frac{\Delta H^2}{\left[\Delta H^2 + \left(H_a - H_R\right)^2\right]}$ and $F_A(H_a) = F_S(H_a)\left[\frac{\left(H_a - H_R\right)}{\Delta H}\right]$ are symmetric and antisymmetric Lorentzian functions, respectively. $S$ and $A$ are the amplitudes of the symmetric $F_S$ and the antisymmetric $F_A$ signals and are proportional to the current-induced in-plane torque ($\tau_\parallel$) and out-of-plane torque ($\tau_\perp$), respectively. Here, $H_a$, $\Delta H$, and $H_R$

refer to the applied external magnetic field, the ferromagnetic resonance linewidth, and the ferromagnetic resonance field, respectively. $H_R$ and $\Delta H$ are extracted and the effective magnetization of the Py layer, $\mu_0 M_{eff}$, is determined by fitting $f$ *vs.* $H_R$ to the Kittel equation, $f = \left(\frac{\gamma}{2\pi}\right)\mu_0\sqrt{\left(H_R - H_k\right)\left(H_R - H_k + M_{eff}\right)}$. The effective Gilbert damping constant $\alpha$ is obtained by a linear fit of $\Delta H$ *vs.* $f$ using $\Delta H = \Delta H_0 + \frac{(2\pi\alpha f)}{\gamma}$[33]. The values for $\mu_0 M_{eff}$ and $\alpha$ for the TaIrTe₄ (90 nm)/Py (6 nm) device are given in Figs. 3a and b, respectively. The $\mu_0 M_{eff}$ and $\alpha$ values for Py, using different thicknesses of TaIrTe₄, are given in Supplementary Fig. 3. The obtained values of the $\mu_0 M_{eff}$ and $\alpha$ are comparable to literature values for Py-5 nm films[34,35].

The strengths of the current-induced torques for different $\phi$ values are related to the symmetries of the device. For example, in conventional Pt/Py bilayers, the two-fold rotational symmetry requires that the SOT changes sign when the magnetization is rotated 180° in-plane, which results in the sign reversal of V_mix while retaining the same amplitude[5]. STFMR measurements with positive ($\phi = 40°$) and negative ($\phi = 220°$) applied fields were performed on the TaIrTe₄/Py devices, and a clear change in both amplitude and shape are obtained as shown in Fig. 2d for the TaIrTe₄(133 nm)/Py(6 nm) device, in the frequency range of 5-7 GHz. The V_mix signal is then fitted with Eq. (1) and the symmetric and antisymmetric components are obtained, which show a large change in amplitude in Fig. 2e for the TaIrTe₄(133 nm)/Py(6 nm)

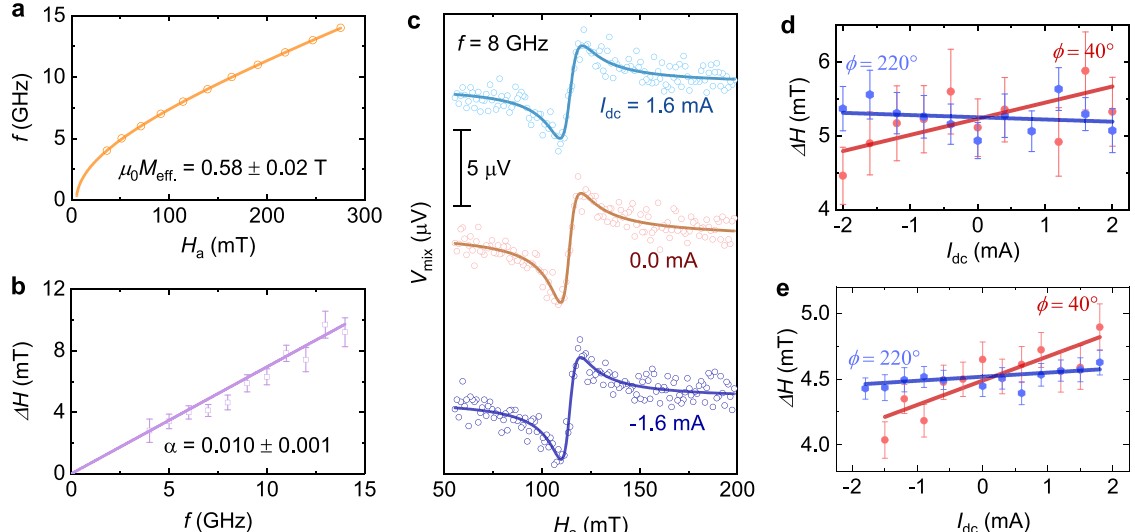

**Fig. 3 | Evaluation of effective spin-orbit torque efficiency from *dc* bias dependence STFMR measurements. a, b** Ferromagnetic resonance frequency *f* vs. resonance field $H_R$ and ferromagnetic resonance linewidth $\Delta H$ vs. *f* for TaIrTe₄(90 nm)/Py(6 nm) device, respectively. **c,** STFMR curves with different values of *dc* current ($I_{dc}$) at 8 GHz, and **d,** $\Delta H$ vs. $I_{dc}$ for TaIrTe₄ (90 nm)/Py (6 nm) device, respectively. **e** $\Delta H$ *versus* $I_{dc}$ for TaIrTe₄(64 nm)/Py (5 nm) device at 7 GHz. In **a–b**, and **d–e**, solid symbols show the extracted values after fitting the experimental data to Eq. 1 and solid lines are fit to the obtained data. While error bars are obtained using the fitting of extracted data to Eq. 1. In **c**, solid symbols show the experimental data points and solid lines are fit to the data using Eq. 1.

device at 5 GHz. It directly indicates that the SOT is affected by the reduced symmetry of the TaIrTe₄ layer and in particular the amplitude difference of the antisymmetric part indicates the clear presence of out-of-plane damping-like torque $\tau_\perp$. The change in amplitude for the symmetric part is also observed for the device measured in Fig. 2e, which indicates the presence of out-of-plane field-like torque in the system[36,37] but it was absent in devices with other thicknesses. The STFMR measurements were also performed on the devices fabricated along the b-axis (current flows along the b-axis) of TaIrTe₄, the representative curves are shown in Fig. 2f. The amplitude of $V_{mix}$ is almost constant in the devices along the b-axis, which confirms that the reduced crystal symmetry along a-axis helps to generate out-of-plane damping-like SOT.

The charge-to-spin conversion efficiency for in-plane ($\sigma_y$) and out-of-plane spin ($\sigma_z$) is evaluated using the symmetric and antisymmetric amplitudes obtained with both positive and negative applied magnetic field $H_a$ at a fixed value of $\phi$ (see Supplementary Note 1 for analysis details)[3,7,38]. The evaluated in-plane damping-like torque ($\xi_{DL,y}$) and out-of-plane damping-like torque ($\xi_{DL,z}$) efficiencies for TaIrTe₄(133 nm)/Py(6 nm) and TaIrTe₄(20 nm)/Py(6 nm) devices are plotted in Supplementary Fig. 4, which show enhancement in SOT efficiency as the $I_{rf}$ frequency is increased. Such a frequency-dependent SOT efficiency has previously been observed in the WTe₂/NiFe system[7]. $\xi_{DL,z}$ varies from 1 to 2.19, while $\xi_{DL,y}$ varies from 0.36 to 0.63 for a frequency range of 5 to 10 GHz. However, as SOT efficiency evaluation using lineshape analysis can be affected by artifact voltages contributing towards $V_{mix}$[32,39,40], we use these analyses to primarily gain a qualitative sense of the in-plane and out-of-plane damping-like SOT components, and then use the more reliable method of *dc* bias linewidth modulation and angular STFMR data for SOT evaluation, as discussed below.

To more accurately characterize the SOT efficiency, *dc* bias-dependent STFMR measurements are done to estimate the effective damping-like torque efficiency[3,32,41], and representative curves for different values of *dc* current ($I_{dc}$) for TaIrTe₄(90 nm)/Py(6 nm) are shown in Fig. 3c. The resonance linewidht, $\Delta H$, is subsequently extracted for different $I_{dc}$ values, and Fig. 3d, e show the resulting $\Delta H$ vs. $I_{dc}$ plots for STFMR devices with TaIrTe₄(90 nm)/Py(6 nm) and TaIrTe₄(64 nm)/

Py(5 nm), respectively. The slope [$\delta\Delta H/\delta(I_{dc})$] of linearly fitted $\Delta H$ vs $I_{dc}$ data indicates the strength of damping-like SOT, and we extract[3,29,32,41],

$$\xi_{DL}^{eff} = \frac{2e}{\hbar} \frac{\left(H_a + 0.5 M_{eff}\right)\mu_0 M_S t_{FM}}{\sin\phi} \frac{\gamma}{2\pi f} \frac{\delta\Delta H}{\delta\left(I_{dc,TaIrTe4}\right)} A_C \quad (2)$$

with $\phi$ the azimuthal angle between $I_{dc}$ and $\mu_0 H_a$, $M_S = 6.4\times10^5$ A/m the saturation magnetization of the Py layer[3,42], $t_{FM}$ the thickness of the Py layer, $\frac{\gamma}{2\pi}$ the effective gyromagnetic ratio of Py, $e$ the elementary charge, $\hbar$ the reduced Planck's constant, $I_{dc,TaIrTe4}$ the current in the TaIrTe₄ layer, and $A_C$ the cross-sectional area of the STFMR microbars. The $I_{dc,TaIrTe4}$ value is estimated by using the measured resistance of the device and the known resistance of the Py layer ($R_{Py} = 329 \, \Omega$)[34,35]. The effective damping-like torque efficiencies of $0.98 \pm 0.21$ and $1.18 \pm 0.54$ are obtained for TaIrTe₄ devices of 64, and 90 nm thicknesses, respectively. The effective spin Hall conductivity ($\sigma_{SHC} = \sigma_c \xi_{DL}^{eff}$) values are evaluated to be $(7.31 \pm 3.30)\times10^4$ $(\hbar/2e)$ $(\Omega m)^{-1}$, $(1.91 \pm 0.42)\times10^4$ $(\hbar/2e)$ $(\Omega m)^{-1}$, and $(-12.86 \pm 3.68)\times10^4$ $(\hbar/2e)$ $(\Omega m)^{-1}$ for TaIrTe₄ device of 90, 64, and 20 nm (see the data in Supplementary Fig. 5), respectively, using the electrical conductivity ($\sigma_c$) values given in Supplementary Table 2 (obtained using the electrical conductivity of Py layer, $\sigma_{Py} = 21.3\times10^5 \Omega^{-1}m^{-1}$). The obtained electrical conductivity values for TaIrTe₄ are in close agreement with the earlier reports[23,26].

As the amplitude of $V_{mix}$ changes significantly with the sign of applied magnetic field in most of the devices (when $\phi$ is varied from 40° to 220°). This behavior of $V_{mix}$ with $\phi$ indicates the presence of unconventional out-of-plane SOT in this system. To analyze this phenomenon in more detail, we performed angle-dependent STFMR measurements, where representative curves for sample TaIrTe₄(120 nm)/Py(6 nm) are shown in Fig. 4a. The S and A values are extracted for different $\phi$ values and their angular dependence is plotted in Fig. 4b, c, respectively. In general, the spin current that generates torque has components along all three *x*, *y* and *z* axes, thus

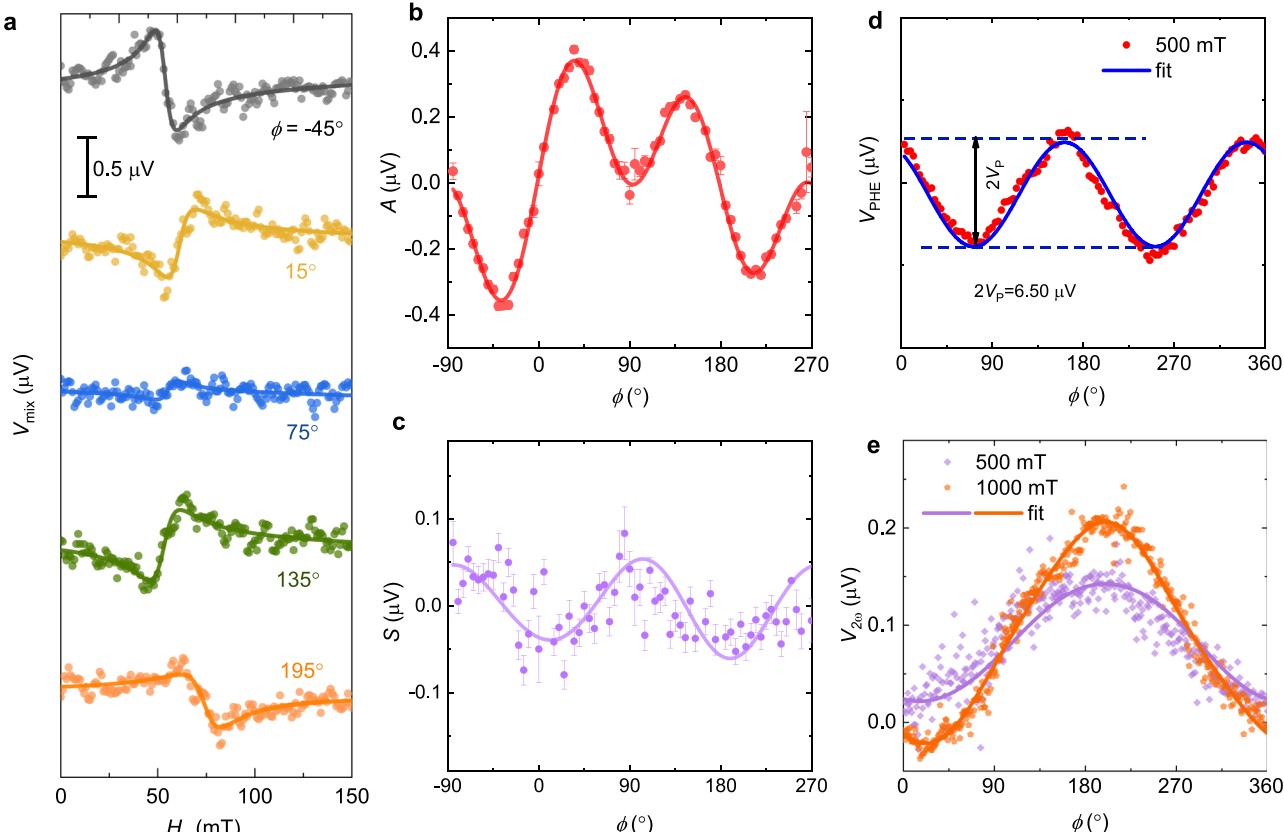

**Fig. 4 | Unconventional spin–orbit torque in TaIrTe₄ from angular STFMR and second harmonic Hall measurements. a** The representative STFMR curves at different values of in-plane magnetic field angle $\phi$ for the frequency of 6 GHz. $\phi$ values are shown in the figure. **b, c** Antisymmetric ($A$) and symmetric ($S$) resonance amplitude versus $\phi$. In both **b** and **c**, solid symbols represent the $A$ and $S$ values extracted after fitting the experimental data to Eq. (1), and solid lines are fit to the obtained data using Eqs. (3) and (4), respectively. Error bar in **b** and **c**, are obtained by fitting the experimental data to Eq. (1). The measurements are performed on the TaIrTe₄(120 nm)/Py(6 nm) STFMR device. **d, e** Planar Hall voltage ($V_{PHE}$) and second harmonic Hall signal ($V_{2\omega}$) as a function of in-plane magnetic field angle, $\phi$, for TaIrTe₄(60 nm)/Py(6 nm) based Hall devices. In **d** and **e**, solid symbols are the experimental data points and solid lines are fit to the data.

the allowed angular dependencies for the coefficients $S$ and $A$ are[3,5,28,43],

$$S = S_{DL}^{Y} \cos \phi \, \sin 2\phi + S_{DL}^{X} \sin \phi \, \sin 2\phi + S_{FL}^{Z} \sin 2\phi \qquad (3)$$

$$A = A_{FL}^{Y} \cos \phi \, \sin 2\phi + A_{FL}^{X} \sin \phi \, \sin 2\phi + A_{DL}^{Z} \sin 2\varphi \qquad (4)$$

where $S_{DL}^{Y}$, $S_{DL}^{X}$, and $A_{DL}^{Z}$ are the coefficients for the damping-like torque with spin polarizations along $x$, $y$, and $z$, respectively; $A_{FL}^{Y}$, $A_{FL}^{X}$, and $S_{FL}^{Z}$ are the corresponding field-like torques. The angular dependence $S$ and $A$ is fitted using Eqs. (3) and (4) and the obtained parameters are given in Supplementary Table 1. While fits of the symmetric component $S$ to Eq. 3 show moderate agreement, the extracted parameters are only required for the $y$- and $x$-polarized spin-current and do not contribute to the $z$-polarized spin-current estimation (see Eqs. 3–5 in Supplementary Note 2).

By considering that $A_{FL}^{Y}$ is due to the Oersted field alone, the amplitudes of the damping-like torque efficiencies per unit current density, $\xi_{DL}^{X}$, $\xi_{DL}^{Y}$, and $\xi_{DL}^{Z}$ in TaIrTe₄ layer can be obtained (see Supplementary Note 2)[3,28]. $\xi_{DL}^{X}$, $\xi_{DL}^{Y}$, and $\xi_{DL}^{Z}$ values of $0.04 \pm 0.01$, $0.08 \pm 0.02$, and $0.11 \pm 0.01$ are obtained for the $t_{TaIrTe4}$ = 120 nm sample. The spin Hall conductivity ($\sigma_{SHC}^{k} = \sigma_c \xi_{DL}^{k}$) can be evaluated using the electrical conductivity ($\sigma_c$) values of TaIrTe₄ (as given in Supplementary Table 1); $\sigma_{SHC}^{Z} = (4.05 \pm 0.23) \times 10^4$ ($\hbar/2e$) ($\Omega$m)⁻¹, $\sigma_{SHC}^{Y} = (2.87 \pm 0.57) \times 10^4$ ($\hbar/2e$) ($\Omega$m)⁻¹, and $\sigma_{SHC}^{X} = (1.43 \pm 0.29) \times 10^4$ ($\hbar/2e$) ($\Omega$m)⁻¹ are obtained for the 120 nm TaIrTe₄ sample. The Dresselhaus-like symmetry torque ($\xi_{DL}^{X}$) is present in this material

system, this type of torque was earlier reported for both TaTe₂/Py and WTe₂/Py heterostructures[17], and it appears due to the Oersted field generated by a component of current flowing transverse to the applied voltage. The in-plane resistivity anisotropy of materials like WTe₂ generates such spatially nonuniform current flow in the heterostructures. To investigate whether non-uniformity in the TaIrTe₄ thickness makes any difference to the SOT efficiency, STFMR devices were made on such flakes (see Supplementary Fig. 6). Interestingly, $\xi_{DL}^{Z}$ shows sign reversal and a value of -0.07 is obtained for this device where the average thickness of TaIrTe₄ is 64 nm.

To further confirm the out-of-plane damping-like SOT efficiency, the harmonic Hall measurements are performed as the STFMR method sometimes gives overestimated values[21]. Low-frequency alternating current ($I_{ac}$) is applied to the devices, and second harmonic Hall voltages as a function of the angle of in-plane magnetic field angle, $\phi$, is measured as shown in Fig. 4e. The damping-like SOT efficiency is estimated using fits of the second harmonic Hall voltage ($V_{2w}$) versus $\phi$, to[5,28,44,45]

$$\begin{aligned} V_{2w} = {}& D_{DL}^{Y} \cos \phi + D_{DL}^{X} \sin \phi + D_{DL}^{Z} \cos 2\phi + F_{FL}^{Y} \cos \phi \, cos 2\phi \\ & + F_{FL}^{X} \sin \phi \, cos 2\phi + F_{FL}^{Z} \end{aligned} \qquad (5)$$

where $D_{DL}^{X}$, $D_{DL}^{Y}$, and $D_{DL}^{Z}$ are the coefficients of damping-like torque with spin polarizations along the $x$, $y$, and $z$, respectively. While, $F_{FL}^{X}$, $F_{FL}^{Y}$, and $F_{FL}^{Z}$ are the field-like torque counterparts. The damping-like torque efficiency is estimated as given in Supplementary Note 3, and

an out-of-plane damping-like torque efficiency ($\theta_{DL}^Z$) of 0.11 ± 0.06 is obtained which is in good agreement with the $\xi_{DL}^Z$ values obtained using the STFMR method.

In summary, we have demonstrated a large unconventional out-of-plane SOT efficiency in the Weyl semimetal candidate TaIrTe$_4$ at room temperature. Using the STFMR and second harmonic Hall measurements of TaIrTe$_4$/Py heterostructure devices, we observed a large damping-like torque due to efficient charge-to-spin conversion. From angular dependent STFMR measurements, the evaluated out-of-plane damping-like torque and the spin Hall conductivity (SHC) $\sigma_{SHC}^Z$ and $\sigma_{SHC}$ values are found to be an *order of magnitude* higher than many transition metal dichalcogenides (TMDs)[13,18,46] including WTe$_2$[5,7], and comparable to the conventional SOT materials such as Bi$_2$Se$_3$[47], and heavy metal Pt[3]. In contrast to conventional materials, TaIrTe$_4$ also exhibits a very large out-of-plane SHC $\sigma_{SHC}^Z$. Utilization of such topological quantum materials with lower crystal symmetries and useful spin textures are therefore suitable for obtaining large charge-to-spin conversion efficiencies with unconventional out-of-plane SOT components. The antisymmetric SOT component in TaIrTe$_4$ has the potential to realize spintronic technologies by enabling efficient manipulation of magnetic materials with perpendicular magnetic anisotropy, leading to high-density, faster, and energy-efficient memory, logic and communication technologies.

*Note*: During the review process of our manuscript, two other papers were also published reporting an out-of-plane SOT in TaIrTe$_4$[48,49].

## Methods
### Single crystal growth
TaIrTe$_4$ crystals were grown by evaporating tellurium from a tellurium-iridium-tantalum melt. The temperature of the melt and crystal growth was 850 °C. The temperature at which tellurium condensed was 720 °C[50]. The chemical composition was studied with a digital scanning electronic microscope TESCAN Vega II XMU with energy dispersive microanalysis system INCA Energy 450/XT (20 kV). The EDX analysis showed that the approximate chemical composition of the samples was close to stoichiometric. No impurity elements and phases were found, see Supplementary Fig. 1 for details.

### Device fabrication
The samples were prepared by mechanically exfoliating nanolayers of TaIrTe$_4$ crystals onto a SiO$_2$/Si wafer by the Scotch tape method inside a glove box. To make TaIrTe$_4$/Ni$_{80}$Fe$_{20}$ heterostructures, the samples were quickly transferred from the glove box to the high vacuum sputtering chamber to deposit Py(5-6 nm)/Al$_2$O$_3$(4 nm) layers. Ni$_{80}$Fe$_{20}$ is known as permalloy (Py). The sample surface was bias sputter cleaned for 20 sec with low energy Ar ion, followed by the deposition of 6 nm Py and 4 nm Al$_2$O$_3$ layers using the dc and rf magnetron sputtering methods, respectively. The TaIrTe$_4$/Py heterostructures were patterned into the rectangular microstrips and Hall devices for spin–torque ferromagnetic resonance (STFMR) and harmonic Hall measurements, respectively, using the electron-beam lithography and Ar ion milling using the negative e-beam resist as the etching mask. Laser writer lithography was used to prepare the top co-planar waveguide contacts for electrical measurements, followed by the lift-off process of 200 nm of copper (Cu) and 20 nm of platinum (Pt).

### Polarized Raman spectroscopy
Polarized, angle-dependent Raman spectra were recorded using an InVia Raman spectrometer from Renishaw. A laser with a wavelength of 785 nm was used as the incident light, using a x50 Leica objective, a 1200 l/mm grating, in the back-scattering mode. The polarization of the laser was rotated from 0° to 180° at angle steps of 10 degrees, while keeping the sample specimen fixed with respect to the

coordinate axes of the laboratory; no analyser was used along the path of the back-scattered light. The recorded Raman spectra were first analyzed as conventional linear plots (intensity vs. Raman shift, as in Fig. 1b) and successfully as polar plots in which the plotted intensities are integrated areas under selected peaks (intensity vs polarization angle, as in Fig. 1c).

### Anisotropic magnetoresistance measurements
In-plane angular dependence anisotropic magnetoresistance (AMR) measurements were performed on ST-FMR microbars using a rotatable projected vector field magnet with a fixed applied magnetic field of 0.1 T and applied dc current of 0.5 mA. The resistance of the devices was measured while rotating the magnetic field in-plane.

### Spin–orbit torque ferromagnetic resonance (STFMR) measurements
STFMR measurements were performed at room temperature on the microbar devices to estimate the spin–orbit torque (SOT) efficiency and other magneto dynamical parameters. The measurements for effective SOT analysis were performed with a fixed in-plane angle $\phi$ = 40°, the radio-frequency (rf) current modulated at 98.76 Hz was applied to the device through a high-frequency bias-T at a fixed frequency (ranging over 3-14 GHz) with an input rf power $P$ = 14 dBm. The in-plane angle $\phi$ dependence STFMR measurements were performed using a rotatable projected field magnet with $\phi$ = 0-360° (step of 5°) at a fixed frequency.

### Harmonic Hall measurements
The schematic of the harmonic measurement setup is given elsewhere[29], where a 213 Hz alternating current ($I_{ac}$) is applied to the channel in the presence of a fixed applied magnetic field, $\mu_0 H_a$. The second harmonic signal is measured while sweeping the in-plane angle, $\phi$, between $\mu_0 H_a$ and $I_{ac}$. Further details of the measurements and analysis are provided in Supplementary Note 3.

## Data availability
The data that support the findings of this study are available from the corresponding authors on request.

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

## Acknowledgements

Authors acknowledge funding from European Union (EU) Graphene Flagship project 2DSPIN-TECH (No. 101135853), EU Graphene Flagship (Core 3, No. 881603), 2D TECH VINNOVA competence center (No. 2019-00068), Swedish Research Council (VR) grant (No. 2021–04821, No. 2018-07046), FLAG-ERA project 2DSOTECH (VR No. 2021-05925), Wallenberg Initiative Materials Science for Sustainability (WISE) funded by the Knut and Alice Wallenberg Foundation, Graphene Center, Chalmers-Max IV collaboration grant, Areas of Advance (AoA) Nano, AoA Materials Science and AoA Energy programs at Chalmers University of Technology. LB acknowledges the support of the Science and Engineering Research Board (SERB) India for the Ramanujan Fellowship. MAH acknowledge support from the VR starting grant 2018-05339 and Wallenberg Foundation (Grant No. 2022.0079). We acknowledge the help of staff at the Quantum Device Physics and Nanofabrication laboratory in our department at Chalmers.

## Author contributions

L.B. and S.P.D. conceived the idea and designed the experiments. L.B., and B.Z. prepared the devices with support from N.B., A.M.H., and L.S. L.B. and N.B. performed STFMR measurements and analysis. B.Z. did the harmonic Hall measurements and L.B. performed the analysis. M.A.H. grew the TaIrTe$_4$ crystals. A.M. performed the Raman experiments. L.B., J.A., and S.P.D. analyzed and interpreted the experimental data and

wrote the manuscript with input from all the authors. S.P.D. coordinated and supervised the project.

## Funding

## Competing interests

The authors declare no competing interests.
