## [Peer Review File · Nature Communications]

Large out-of-plane spin-orbit torque in topological Weyl semimetal TaIrTe₄Reviewers' Comments:

Reviewer #1:

Remarks to the Author:

The work by Bainsla et al. reports the large unconventional spin polarization generated by TaIrTe₄. The authors quantified the spin torque efficiencies associated with different spin polarization using angular-dependent ST-FMR. The reported out-of-plane spin hall conductivity ($4.05 \pm 104 \text{ Ohm}^{-1} \text{ m}^{-1}$) is high, but the study lacks strong experimental evidences and systematical study for unconventional spin generation, reliable SOT characterization and data, and device demonstration. Overall, this work is not suitable to publish in its present form. Below please see my detailed comments:

1. In the introduction section, the background investigation is not sufficient. Wely semimetal, eg., WTe₂, is only a small part of the promising unconventional spin materials; some other materials, such as topological materials, AFM materials, FM materials, et al. could present very significant unconventional spin efficiency, and also demonstrate field-free switching. Please include more recently published materials as comparison in the background, and indicate why TaIrTe₄ as investigated in this work is a better option.
2. The authors claim that the low crystal symmetry of Td-TaIrTe₄ provides unconventional spin generation. However, no experimental evidence is shown about the details of the crystalline structure on the films, and no explanation that certain crystallinity is responsible for the significant unconventional spins. The EDX image shown in supplementary info doesn't provide enough evidence.
3. As authors pointed out in page6, the ST-FMR method using lineshape analysis is not accurate, and all the related data is just preliminary results. I think this part is not suitable to be presented in the manuscript.
4. For the DC bias dependent ST-FMR measurement, I have two concerns here: 1) The error bars in Fig. 3d, 3e are quite significant and the accuracy of this linear fitting is not acceptable to be published in Nat. Comm. Authors could try a higher RF source power or a different modulation frequency for better signals. 2) Angular-dependent ST-FMR is a good way to characterize SOT, however, it shows quite different results from SHH method, and might overestimate the efficiency, see reference "X. Chen et al, Nat. Mat. 20, 800-804 (2021)". Please comment on this, and try independent SOT characterization methods. A control sample will be helpful as well.
5. Some equations are not necessarily needed to be in the main manuscript, e.g., Eq. (5)-(7).
6. The spin-orbital coupling material study is not systematic with regard to performances (SOT efficiency, resistivity, etc.) dependence on thickness, temperature, etc. I understand that for the film achieved by mechanical exfoliation, it is not easy to control the thickness, but SOT efficiency at different thickness will comprehensively help to depict the spin transport behavior in this material.
7. The authors didn't show any device demonstration with field-free switching, which could be a strong evidence of existence of unconventional spins. There might be difficulty growing perpendicular FL on the exfoliated TaIrTe₄ for device demonstration, this work is still needed.

Reviewer #2:

Remarks to the Author:

The type II Weyl semimetals have been widely studied for generating unconventional spin orbit torques due to the breaking of crystal mirror symmetry. Besides extensively explored Td-WTe₂, TaIrTe₄ is another kind of type II Weyl semimetals. The authors investigated the spin-orbit torque and spin conductivity of TaIrTe₄ with varying thicknesses using the standard ST-FMR method. Their main claim is that TaIrTe₄ has a higher out-of-plane spin Hall conductivity, which is an order higher than the reported values in other materials including WTe₂.

In current work, the study of spin-orbit torque and spin conductivity in TaIrTe₄ is not systematic. The SOT efficiency and spin conductivity for different thicknesses has not been thoroughly extracted. Particularly, the authors do not present sufficient evidence to support their primary claim of a large out-of-plane spin conductivity in TaIrTe₄. From the perspective of unconventional spin generation

mechanism, there is no fundamental difference between WTe₂ and TaIrTe₄. In this regard, I do not find the comprehensiveness and the level of novelty in the present work meet the high standards for publication in Nature Communications.

Major issues:

1. In page 2, line 5-7 of the main text, the statement 'Furthermore, as the component of torque lies in-plane in conventional SOMs, they are only suitable for deterministic switching of in-plane magnets' is erroneous. In conventional SOMs, the orientations of torques dependent on the magnetization direction.
2. In page 3, line 1-2 of the main text, the statement 'a large out-of-plane SOT, a large out-of-plane damping-like torque component, and a large spin Hall conductivity' is confusing. Out-of-plane SOT should include the out-of-plane damping-like torque component. The authors should also specify the spin Hall conductivity is in-plane or out-of-plane. Actually through the whole manuscript, the authors did not clearly differentiate SOT from damping-like SOT and field-like SOT, which weakens the readability of this manuscript.
3. The author should confirm the crystal axis orientation of their devices by direct material characterization methods such as polarized Raman spectroscopy, instead of just by observing the longer axis of the exfoliated flake.
4. To validate that the unconventional spin arises from reduced crystal-symmetry rather than other factors like non-uniformity or strain effect, it is suggested that the authors provide a comparison of ST-FMR results for the current along both the a and b axes.
5. In the supplementary Table 1, it is noted that for thicker TaIrTe₄ samples, the SOT efficiency of σ_z increases slightly (from 0.07 to 0.11) while the SOT efficiency of σ_y decreases sharply (0.97 to 0.08). It is suggested that the authors conduct ST-FMR measurements for the samples with various TaIrTe₄ thicknesses to provide a detailed thickness-dependent trend analysis.
6. For the lattice structure, TaIrTe₄ closely resembles WTe₂, both preserving mirror symmetry along only one axis within the xy plane. Notably, this paper reports a decent Dresselhaus torque (σ_x) in TaIrTe₄, which is rarely reported in WTe₂ by using ST-FMR. The authors need to provide more comprehensive discussions and explanations regarding σ_x .
7. What is the relationship between the spin conductivities obtained in page 7 and page 9? In fact, if there is out-of-plane spin induced torque, the dc biased ST-FMR evaluation equation should be changed accordingly. I suggest the authors extract the out-of-plane spin conductivity in a thickness-dependent manner and give a discussion.
8. In the last sentence of page 7, the authors should be more explicit regarding the charge conductivity of TaIrTe₄ with different thicknesses used to evaluate the spin conductivity.
9. The authors should measure and state the TaIrTe₄ resistivity and Py resistivity used in their experiments. Whether the resistivities are comparable with reported value?
10. In supplementary figure 2, the AMR was performed on TaIrTe₄(64 nm)/Py(6nm), but the ST-FMR measurement was performed on TaIrTe₄(64 nm)/Py(5nm). The thicknesses of Py are inconsistent. Actually, the AMR must obtain from the same device in ST-FMR measurement for the accurate extraction of SOTs or spin conductivities. In fact, I don't see anywhere the AMR value is used in their analysis. Why do the authors provide these data?

Other comments:

- in the last sentence of the 1st paragraph in page 6, the authors should specify the type of the out-of-plane torque. Is it damping-like or field-like? Otherwise, this statement is incorrect.
- there is a clear amplitude difference of SFs for positive and negative fields in Fig. 2e. It seems there is out-of-plane spin induced field-like torque. The author should add a discussion on this.
- in page 9, all three spin conductivity should have error bar.
- There are typos in page 5 line 6. HR and ΔH are not magnetization of the Py layer.

Reviewer #3:

Remarks to the Author:

This work presents a study on a large out-of-plane damping-like SOT at room temperature using a topological Weyl semimetal candidate TaIrTe₄ with a lower crystal symmetry. It's intriguing that the study finds a larger out-of-plane spin Hall conductivity, which is one order larger than other systems. This allows for field-free switching of perpendicular magnetization with higher efficiency. However, I think there are some issues mentioned below; if they are properly addressed in the revised manuscript, I would consider recommending the publication of this manuscript in Nature Communications.

1. It would be perfect if the authors could demonstrate the perpendicular magnetization switching using the large out-of-plane damping-like torque SOT. Then they can check if the threshold current density is indeed lower as compared with other systems.
2. All the claims in this paper rely on the ST-FMR methods. The author may consider alternative approach for analyzing the symmetries of their torques, such as second harmonic Hall technique. In addition, the authors could use a dc-bias current modulated ST-FMR approach for identifying different damping-like torque contributions.
3. The author may add the ST-FMR data from the devices fabricated along b-axis of TaIrTe₄ to verify that the out-of-plane damping-like torque is from the low crystal symmetry origin.

Date: 21/03/2024

Rebuttal on the manuscript “Large out-of-plane spin-orbit torque in topological Weyl semimetal TaIrTe₄” submitted to Nature Communications.

We are very grateful for handling of our manuscript and for the thorough reports by the editor and all the reviewers. The reviewers raised several important questions, which we feel we have now addressed in this rebuttal and our new version of the manuscript. We thank the referees for their careful reading of our manuscript, which we feel has greatly improved the manuscript. Thanks to their valuable insight and we hope that they find that our manuscript to be suitable for publication in Nature Communications.

Please find below a point-by-point response to all questions raised. For additional clarity, all changes to the manuscript have been marked in red.

REVIEWER COMMENTS

Reviewer #1

The work by Bainsla et al. reports the large unconventional spin polarization generated by TaIrTe₄. The authors quantified the spin torque efficiencies associated with different spin polarization using angular-dependent ST-FMR. The reported out-of-plane spin hall conductivity ($4.05 \pm 10^4 \text{ Ohm}^{-1} \text{ m}^{-1}$) is high, but the study lacks strong experimental evidences and systematical study for unconventional spin generation, reliable SOT characterization and data, and device demonstration. Overall, this work is not suitable to publish in its present form. Below please see my detailed comments:

Response: We express our gratitude to the reviewer for their invaluable feedback. We have meticulously examined all the comments provided by the reviewer and diligently endeavored to address each concern raised. We are confident that the revisions made to our manuscript have significantly enhanced its quality, rendering it well-suited for publication in Nature Communications. Below, we present a detailed point-by-point response to the reviewer's remarks.

Comment 1: In the introduction section, the background investigation is not sufficient. Wely semimetal, eg., WTe₂, is only a small part of the promising unconventional spin materials; some other materials, such as topological materials, AFM materials, FM materials, et al. could present very significant unconventional spin efficiency, and also demonstrate field-free switching. Please include more recently published materials as comparison in the background, and indicate why TaIrTe₄ as investigated in this work is a better option.

Response 1: We express our appreciation to the reviewer for highlighting this crucial aspect. In response, we have expanded the scope of materials included for comparison within the

manuscript. It is noteworthy that thus far, the unconventional spin-orbit torque (SOT) exhibited by various material classes has typically been an order of magnitude smaller than conventional torques. Therefore, the quest for identifying novel materials with enhanced unconventional SOT holds paramount significance. In our study, we have successfully demonstrated the attainment of substantial unconventional SOT in TaIrTe₄ through comprehensive spin-transfer ferromagnetic resonance (STFMR) measurements. Furthermore, to support our findings, we have conducted harmonic Hall measurements, the results of which have been incorporated into the revised manuscript.

Comment 2: The authors claim that the low crystal symmetry of Td-TaIrTe₄ provides unconventional spin generation. However, no experimental evidence is shown about the details of the crystalline structure on the films, and no explanation that certain crystallinity is responsible for the significant unconventional spins. The EDX image shown in supplementary info doesn't provide enough evidence.

Response 2: We have conducted Energy Dispersive X-ray (EDX) measurements on the bulk crystals to validate the composition of TaIrTe₄ utilized in our study. The results affirm the near-perfect composition of our crystals, with any deviation falling within the one percent margin of error typically associated with EDX analysis, and it has been found that it results in a Td-TaIrTe₄ crystal structure (please see reference 21 in the manuscript).

Moreover, in the revised manuscript, we have incorporated polarized Raman spectroscopy measurements to ascertain the orientation of TaIrTe₄ flakes. Our findings reveal that the longer axis of the flakes aligns with the a-axis, while the shorter axis corresponds to the b-axis. Additionally, we have fabricated STFMR devices oriented along both the a and b-axes. Intriguingly, our experiments demonstrate the presence of unconventional spin-orbit torque (SOT) exclusively in devices aligned along the a-axis, while devices fabricated along the b-axis exhibit no discernible evidence of unconventional SOT.

Comment 3: As authors pointed out in page6, the ST-FMR method using lineshape analysis is not accurate, and all the related data is just preliminary results. I think this part is not suitable to be presented in the manuscript.

Response 3: We acknowledge the reviewer's apprehension regarding lineshape analysis. It is imperative to clarify that our utilization of these analyses in the manuscript was primarily qualitative, aiming to discuss the efficiency of SOT without claiming quantitative precision. Subsequently, to substantiate our findings, we employed additional characterization techniques such as angular STFMR to derive more precise SOT efficiency values.

In response to the concerns raised, we have relocated the lineshape analysis figure (Supplementary Fig. 4) to the Supplementary Information section of the revised manuscript. This adjustment ensures a clearer delineation between qualitative and quantitative assessments of SOT efficiency, thereby enhancing the clarity of our findings.

Comment 4: For the DC bias dependent ST-FMR measurement, I have two concerns here: 1) The error bars in Fig. 3d, 3e are quite significant and the accuracy of this linear fitting is not acceptable to be published in Nat. Comm. Authors could try a higher RF source power or a

different modulation frequency for better signals. 2) Angular-dependent ST-FMR is a good way to characterize SOT, however, it shows quite different results from SHH method, and might overestimate the efficiency, see reference “X. Chen et al, Nat. Mat. 20, 800-804 (2021)”. Please comment on this, and try independent SOT characterization methods. A control sample will be helpful as well.

Response 4: We express our gratitude to the reviewer for acknowledging the utility of angular STFMR as a valuable method for characterizing SOT. We agree with the reviewer's observation regarding the significant error bars depicted in Fig. 3d and 3e of our manuscript.

In response to the reviewer's suggestion, we have already made careful efforts to enhance the quality of our data by varying the RF power and modulation frequency. The results of these optimizations have been incorporated into Fig. 3d and 3e, showcasing our most refined dc bias-dependent measurements. It is important to emphasize that these dc bias-dependent measurements were instrumental in estimating the effective damping-like torque, also referred to as effective SOT.

Furthermore, to strengthen our findings, we have undertaken additional steps in the revised manuscript. Specifically, we have fabricated Hall devices and conducted second harmonic Hall measurements to accurately estimate the z-polarized damping-like torque. Notably, the values obtained through harmonic Hall measurements exhibit comparability with those derived from angular STFMR data. This alignment between the two sets of measurements lends robust support to our assertion regarding the substantial damping-like out-of-plane spin Hall conductivity inherent in TaIrTe₄.

Comment 5: Some equations are not necessarily needed to be in the main manuscript, e.g., Eq. (5)-(7).

Response 5: We appreciate the reviewer's suggestion, and in line with ensuring clarity and conciseness in the main manuscript, we have relocated equations (5)-(7) to the Supplementary information section. A reference to this Supplementary information has been included in the main manuscript for easy access by interested readers.

Additionally, to provide detailed insight into the estimation process of damping-like torque efficiencies with spin polarizations along the x, y, and z directions, we have introduced a new supplementary note titled "Supplementary Note 2. Estimation of damping-like torque efficiencies with spin polarizations along x, y, and z directions". This supplementary note elucidates the methodology employed in estimating these torque efficiencies, thereby enhancing the comprehensibility and transparency of our research findings.

Comment 6: The spin-orbital coupling material study is not systematic with regard to performances (SOT efficiency, resistivity, etc.) dependence on thickness, temperature, etc. I understand that for the film achieved by mechanical exfoliation, it is not easy to control the thickness, but SOT efficiency at different thickness will comprehensively help to depict the spin transport behavior in this material.

Response 6: We appreciate the reviewer's acknowledgment of the challenges associated with controlling TaIrTe₄ thickness through mechanical exfoliation. In response, we have undertaken the fabrication of new Hall and STFM devices, and have incorporated the corresponding data into the revised manuscript. This addition serves to enhance the comprehensiveness and robustness of our findings.

Furthermore, we have conducted second harmonic Hall measurements to ascertain the damping-like SOT efficiency, with results demonstrating good agreement with values estimated through the STFM method. This alignment supports the reliability of our measurements.

Moreover, to provide further insights into the material properties, we have included data on the electrical conductivity at various thicknesses of TaIrTe₄ in the supplementary information. This supplemental information improves the characterization of the material and facilitates a comprehensive understanding of its behavior.

It is imperative to note that our focus lies on the room-temperature application of the material. Hence, all measurements and analyses have been carefully conducted exclusively at room temperature, ensuring relevance to real-world applications under typical operating conditions.

Comment 7: The authors didn't show any device demonstration with field-free switching, which could be a strong evidence of existence of unconventional spins. There might be difficulty growing perpendicular FL on the exfoliated TaIrTe₄ for device demonstration, this work is still needed.

Response 7: We appreciate the reviewer's insight regarding the significance of demonstrating field-free switching as strong evidence for the presence of unconventional torques. We acknowledge the challenges associated with growing perpendicular magnetic layers over TaIrTe₄, which indeed requires substantial time and effort. In our commitment to advancing the understanding of unconventional torques in TaIrTe₄, we have noted the reviewer's suggestion for future studies.

Reviewer #2

Comment: The type II Weyl semimetals have been widely studied for generating unconventional spin orbit torques due to the breaking of crystal mirror symmetry. Besides extensively explored Td-WTe₂, TaIrTe₄ is another kind of type II Weyl semimetals. The authors investigated the spin-orbit torque and spin conductivity of TaIrTe₄ with varying thicknesses using the standard ST-FMR method. Their main claim is that TaIrTe₄ has a higher out-of-plane spin Hall conductivity, which is an order higher than the reported values in other materials including WTe₂.

In current work, the study of spin-orbit torque and spin conductivity in TaIrTe₄ is not systematic. The SOT efficiency and spin conductivity for different thicknesses has not been

thoroughly extracted. Particularly, the authors do not present sufficient evidence to support their primary claim of a large out-of-plane spin conductivity in TaIrTe₄. From the perspective of unconventional spin generation mechanism, there is no fundamental difference between WTe₂ and TaIrTe₄. In this regard, I do not find the comprehensiveness and the level of novelty in the present work meet the high standards for publication in Nature Communications.

Response: We extend our gratitude to the reviewer for dedicating their time to provide valuable insights and suggestions to enhance the quality of our research. As pointed out by the reviewer, the mechanisms underlying the unconventional SOT generation in both WTe₂ and TaIrTe₄ are fundamentally similar. However, it's noteworthy that the damping-like out-of-plane spin Hall conductivity of TaIrTe₄ surpasses that of other materials, including WTe₂, by an order of magnitude. This attribute holds significant promise for minimizing power consumption in spintronic devices.

To substantiate our claims regarding the substantial damping-like out-of-plane spin Hall conductivity in TaIrTe₄, we conducted experiments involving the fabrication of new Hall and STFM devices, yielding results that corroborate our assertions. Additionally, polarized Raman spectroscopy measurements were employed to ascertain the orientation of our flakes along the a and b axes, further reinforcing our contention that the longer axis corresponds to the a-axis.

We are confident that the incorporation of these new findings and analyses has elevated the quality of our manuscript, aligning it with the rigorous standards set by Nature Communications for publication. We meticulously reviewed and addressed each of the comments provided by the reviewers to the best of our abilities.

Major issues:

Comment 1: In page2, line5-7 of the main text, the statement ‘Furthermore, as the component of torque lies inplane in conventional SOMs, they are only suitable for deterministic switching of in-plane magnets’ is erroneous. In conventional SOMs, the orientations of torques dependent on the magnetization direction.

Response 1: We appreciate the helpful suggestion from the reviewer. We have updated the text to reflect the following: "Moreover, since the torque component predominantly lies in-plane in conventional spin-orbit materials (SOMs) like Pt and Ta, they are primarily suited for deterministic switching of in-plane magnets."

Comment 2: In page3, line1-2 of the main text, the statement ‘a large out-of-plane SOT, a large out-of-plane damping-like torque component, and a large spin Hall conductivity’ is confusing. Out-of-plane SOT should include the out-of-plane damping-like torque component. The authors should also specify the spin Hall conductivity is in-plane or out-of-plane. Actually through the whole manuscript, the authors did not clearly differentiate SOT from damping-like SOT and field-like SOT, which weakens the readability of this manuscript.

Response 2: We are grateful to the reviewer for highlighting this crucial point. We have streamlined the text to state: "We showcase a significant out-of-plane damping-like torque and

a substantial out-of-plane spin Hall conductivity at room temperature." It's important to note that our manuscript primarily focuses on estimating the damping-like SOT, and as such, does not delve into discussions regarding the field-like SOT.

Comment 3: The author should confirm the crystal axis orientation of their devices by direct material characterization methods such as polarized Raman spectroscopy, instead of just by observing the longer axis of the exfoliated flake.

Response 3: We express our gratitude to the reviewer for providing this insightful comment, which has contributed to the enhancement of our manuscript. In response, we conducted polarized Raman spectroscopy measurements on our samples, confirming that the longer axis in our flakes aligns with the a-axis. We have incorporated the polarized Raman spectroscopy data into Figure 1 of the main manuscript and added relevant discussions regarding these findings.

Comment 4: To validate that the unconventional spin arises from reduced crystal-symmetry rather than other factors like non-uniformity or strain effect, it is suggested that the authors provide a comparison of ST-FMR results for the current along both the a and b axes.

Response 4: This is a highly valuable suggestion, which prompted us to verify the origin of unconventional torques. To address this, we fabricated and measured new devices with current flowing separately along the a and b axes. Our observations revealed the presence of unconventional torques only when the current flows along the a-axis, whereas no such torques were observed when the current is directed along the b-axis. We have included the data for devices with current flowing along the b-axis in Figure 2 for comparative analysis.

Comment 5: In the supplementary Table1, it is noted that for thicker TaIrTe4 samples, the SOT efficiency of σ_z increases slightly (from 0.07 to 0.11) while the SOT efficiency of σ_y decreases sharply (0.97 to 0.08). It is suggested that the authors conduct ST-FMR measurements for the samples with various TaIrTe4 thicknesses to provide a detailed thickness-dependent trend analysis.

Response 5: We acknowledge the reviewer's suggestion regarding the need for additional STFMR data to better capture trends in σ_z and σ_y . Consequently, we fabricated new devices and conducted both STFMR and harmonic Hall measurements on them. While we were unable to perform additional angular-dependent STFMR measurements, we have included new dc bias-dependent STFMR data (supplementary Fig. 5) and derived effective damping-like SOT values as presented in the revised manuscript. Additionally, we carried out second harmonic Hall measurements (Fig. 4d and e), and the obtained out-of-plane damping-like SOT values are in satisfactory agreement with those estimated using the STFMR method.

Comment 6: For the lattice structure, TaIrTe4 closely resembles WTe2, both preserving mirror symmetry along only one axis within the xy plane. Notably, this paper reports a decent Dresselhaus torque (σ_x) in TaIrTe4, which is rarely reported in WTe2 by using ST-FMR. The authors need to provide more comprehensive discussions and explanations regarding σ_x .

Response 6: The Dresselhaus-like symmetry torque (ξ_{DL}^X) has been previously reported in TaTe₂/Py and WTe₂/Py heterostructures (referenced in our manuscript, see reference 17), and

it arises from the Oersted field generated by a component of current flowing perpendicular to the applied voltage. This phenomenon is induced by the in-plane resistivity anisotropy observed in materials like WTe₂, leading to spatially nonuniform current flow within the heterostructures. We have incorporated this discussion into the manuscript.

Comment 7: What is the relationship between the spin conductivities obtained in page 7 and page 9? In fact, if there is out-of-plane spin induced torque, the dc biased ST-FMR evaluation equation should be changed accordingly. I suggest the authors extract the out-of-plane spin conductivity in a thickness-dependent manner and give a discussion.

Response 7: The spin Hall conductivities (σ_{SHC}) discussed on page 7 of the manuscript represent the effective spin Hall conductivity, which is derived from the effective damping-like SOT efficiency (ξ_{DL}^{eff}) and the longitudinal charge conductivity (σ_c) values ($\sigma_{SHC} = \sigma_c \xi_{DL}^{eff}$). Meanwhile, the spin Hall conductivities mentioned on page 9 are the spin Hall conductivities ($\sigma_{SHC}^k = \sigma_c \xi_{DL}^k$) associated with spin polarization along the k -axis, where $k = x, y, z$.

Comment 8: In the last sentence of page 7, the authors should be more explicit regarding the charge conductivity of TaIrTe₄ with different thicknesses used to evaluate the spin conductivity.

Response 8: We have included a table detailing the electrical charge conductivity versus thickness of TaIrTe₄ in the supplementary information file, as well as mentioned in the main manuscript.

Comment 9: The authors should measure and state the TaIrTe₄ resistivity and Py resistivity used in their experiments. Whether the resistivities are comparable with reported value?

Response 9: We have now included the electrical conductivity table for TaIrTe₄ in the Supplementary information file. Additionally, the electrical resistance (which allows calculation of resistivity by considering device dimensions) for Py was already provided in the main manuscript. The electrical conductivity values obtained for TaIrTe₄ in our study are consistent with earlier reports (referenced in our manuscript, see references 23 and 26).

Comment 10: In supplementary figure2, the AMR was performed on TaIrTe₄(64 nm)/Py(6nm), but the STFMR measurement was performed on TaIrTe₄(64 nm)/Py(5nm). The thicknesses of Py are inconsistent. Actually, the AMR must obtain from the same device in STFMR measurement for the accurate extraction of SOTs or spin conductivities. In fact, I don't see anywhere the AMR value is used in their analysis. Why do the authors provide these data?

Response 10: We appreciate the reviewer for identifying the inconsistency in the Py thickness. Indeed, it was a typing error in Supplementary figure 2, and the devices have the same Py thickness. Regarding the use of AMR values, they are not employed to extract SOT values. However, we checked the AMR values to ensure the quality of our devices. This aspect is also mentioned in the main manuscript:

“In STFM measurements, an in-plane radio frequency current I_{rf} is applied along the a-axis of TaIrTe₄ while an in-plane magnetic field is applied at an angle ϕ with respect to the I_{rf} , as shown in Fig. 2a. I_{rf} in TaIrTe₄ generates a spin current in the z-direction, which is injected into the adjacent Py layer and excites the Py into a precessional motion. Thanks to its anisotropic magnetoresistance (AMR), the resistance of Py oscillates with the same frequency as that of I_{rf} , and produces a *dc* mixing voltage V_{mix} , which is then measured using a lock-in amplifier.”

Other comments:

Comment 11: in the last sentence of the 1st paragraph in page 6, the authors should specify the type of the out-of-plane torque. Is it damping-like or field-like? Otherwise, this statement is incorrect.

Response 11: The amplitude difference observed in the antisymmetric part of V_{mix} clearly indicates the presence of out-of-plane damping-like torque. We have incorporated this clarification into the main manuscript.

Comment 12: there is a clear amplitude difference of SFs for positive and negative fields in Fig. 2e. It seems there is out-of-plane spin induced field-like torque. The author should add a discussion on this.

Response 12: Yes, the change in amplitude for the symmetric part for the device measured in Fig. 2e, indicates the presence of out-of-plane field-like torque in the system (see references 36 and 37 of the manuscript) but it was absent in devices with other thicknesses. Therefore, we refrained from including any analysis or discussion on out-of-plane field-like torque in the manuscript. We have mentioned this in our revised manuscript.

Comment 13: in page9, all three spin conductivity should have error bar.

Response 13: In the revised manuscript, we have included the error bar in all three spin conductivities.

Comment 14: There are typos in page 5 line 6. H_R and ΔH are not magnetization of the Py layer.

Response 14: We have reviewed the text and confirmed that H_R refers to the ferromagnetic resonance field, while ΔH refers to the ferromagnetic resonance linewidth. This information is accurately reflected in the main manuscript:

“Here, H_a , ΔH , and H_R refer to the applied external magnetic field, the ferromagnetic resonance linewidth, and the ferromagnetic resonance field, respectively. H_R and ΔH are extracted and the effective magnetization of the Py layer, $\mu_0 M_{eff}$, is determined by fitting f vs.

H_R to the Kittel equation, $f = \left(\frac{\gamma}{2\pi}\right) \mu_0 \sqrt{(H_R - H_k)(H_R - H_k + M_{eff})}$.”

Reviewer #3

This work presents a study on a large out-of-plane damping-like SOT at room temperature using a topological Weyl semimetal candidate TaIrTe₄ with a lower crystal symmetry. It's intriguing that the study finds a larger out-of-plane spin Hall conductivity, which is one order larger than other systems. This allows for field-free switching of perpendicular magnetization with higher efficiency. However, I think there are some issues mentioned below; if they are properly addressed in the revised manuscript, I would consider recommending the publication of this manuscript in Nature Communications.

We express our gratitude to the reviewer for their valuable comments and for considering our work suitable for Nature Communications after the revision. We have thoroughly examined all the reviewer's comments and made diligent efforts to address each of them. Below is our point-by-point response to the reviewer's comments:

Comment 1: It would be perfect if the authors could demonstrate the perpendicular magnetization switching using the large out-of-plane damping-like torque SOT. Then they can check if the threshold current density is indeed lower as compared with other systems.

Response 1: We acknowledge the reviewer's suggestion regarding the potential improvement in the quality of our work through the demonstration of perpendicular magnetization switching. However, achieving perpendicular magnetization on TaIrTe₄ requires a significant amount of time and effort, which we plan to explore in future studies.

Comment 2: All the claims in this paper rely on the ST-FMR methods. The author may consider alternative approach for analyzing the symmetries of their torques, such as second harmonic Hall technique. In addition, the authors could use a dc-bias current modulated ST-FMR approach for identifying different damping-like torque contributions.

Response 2: We express our gratitude to the reviewer for the suggestion. In response, we have fabricated new Hall devices and conducted second harmonic Hall measurements and analysis. The out-of-plane damping-like SOT efficiency obtained using the second harmonic method agrees with the values obtained using the STFMR method.

Comment 3: The author may add the ST-FMR data from the devices fabricated along b-axis of TaIrTe₄ to verify that the out-of-plane damping-like torque is from the low crystal symmetry origin.

Response 3: We extend our appreciation to the reviewer for this valuable suggestion to verify the origin of the out-of-plane damping-like torque. In response, we fabricated and measured devices with current flowing along the b-axis. We observed a symmetric V_{mix} signal for these devices, confirming the absence of unconventional torque. The unconventional out-of-plane damping-like torque is only observed when the current flows along the a-axis.

Reviewers' Comments:

Reviewer #1:

Remarks to the Author:

The revisions and the rebuttal are satisfactory at this round.

I can accept the manuscript for publication if new concerns from other reviewers are addressed.

Reviewer #2:

Remarks to the Author:

The authors have addressed the most questions and the manuscript can be accepted now.

Reviewer #3:

Remarks to the Author:

The authors have answered the points raised, and the manuscript has been sufficiently improved to recommend publication.